# Interacting active surfaces: A model for three-dimensional cell aggregates

**Alejandro Torres-Sánchez**  [1¤], **Max Kerr Winter** [1], **Guillaume Salbreux** [1,2]*

**1** Theoretical Physics of Biology laboratory, The Francis Crick Institute, London, United Kingdom,
**2** Department of Genetics and Evolution, University of Geneva, Genève, Switzerland

¤ Current address: Tissue Biology and Disease Modelling Unit, European Molecular Biology Laboratory, Barcelona, Spain
* guillaume.salbreux@unige.ch

## Abstract

We introduce a modelling and simulation framework for cell aggregates in three dimensions based on interacting active surfaces. Cell mechanics is captured by a physical description of the acto-myosin cortex that includes cortical flows, viscous forces, active tensions, and bending moments. Cells interact with each other via short-range forces capturing the effect of adhesion molecules. We discretise the model equations using a finite element method, and provide a parallel implementation in C++. We discuss examples of application of this framework to small and medium-sized aggregates: we consider the shape and dynamics of a cell doublet, a planar cell sheet, and a growing cell aggregate. This framework opens the door to the systematic exploration of the cell to tissue-scale mechanics of cell aggregates, which plays a key role in the morphogenesis of embryos and organoids.

## Author summary

Understanding how tissue-scale morphogenesis arises from cell mechanics and cell-cell interactions is a fundamental question in developmental biology. Here we propose a mathematical and numerical framework to address this question. In this framework, each cell is described as an active surface representing the cell acto-myosin cortex, subjected to flows and shape changes according to active tensions, and to interaction with neighbouring cells in the tissue. Our method accounts for cellular processes such as cortical flows, cell adhesion, and cell shape changes in a deforming three-dimensional cell aggregate. To solve the equations numerically, we employ a finite element discretisation, which allows us to solve for flows and cell shape changes with arbitrary resolution. We discuss applications of our framework to describe cell-cell adhesion in doublets, three-dimensional cell shape in a simple epithelium, and three-dimensional growth of a cell aggregate.

## 1 Introduction

Tissue morphogenesis relies on the controlled generation of the cellular forces that collectively drive tissue-scale flows and deformation [1, 2]. The interplay between cell-cell adhesion,

**Data Availability Statement:** The source code used to produce the results and analyses presented in this manuscript is available at the Git repository: https://github.com/torressancheza/ias.

**Funding:** ATS, MKW and GS acknowledge support from the Francis Crick Institute, which receives its core funding from Cancer Research UK (FC001317, https://www.cancerresearchuk.org), the UK Medical Research Council (FC001317, https://www.ukri.org/councils/mrc/), and the Wellcome Trust (FC001317, https://wellcome.org). GS and ATS acknowledge support from a grant to GS, Chris Dunsby, Axel Behrens from the Engineering and Physical Sciences Research Council (EP/T003103/1, https://epsrc.ukri.org). The funders had no role in study design, data collection and analysis, decision to publish, or preparation of the manuscript.

**Competing interests:** The authors have declared that no competing interests exist.

cellular mechanics and the cytoskeleton plays a key role in determining how biological tissues self-organise [3]. These ingredients are also crucial for the growth of *in vitro* organoids, organ-like systems derived from stem cells which can self-organise into complex structures reminiscent of actual organs [4–6].

Several classes of models have been proposed to describe the mechanics of multicellular aggregates, such as cellular Potts models [7–13], phase field models [14–18] and vertex models [19]. Vertex models, and the closely related Voronoi vertex models [20, 21], describe cells in a tissue as polyhedra that share faces, edges and vertices forming a three-dimensional junctional network [19, 22–24]. Cell deformations are encoded by the displacement of the vertices $X_a$ of the network. These displacements are dictated by the balance of vertex forces $F_a$ stemming from cell pressures $P_c$, surface tensions $t_f$ and line tensions $\Gamma_e$ that are coupled to virtual changes in cell volume $\delta V_c$, face area $\delta A_f$ and edge length $\delta l_e$ respectively in a work function

$$\delta W = \sum_{c \in \text{cells}} - P_c \delta V_c + \sum_{f \in \text{faces}} t_f \delta A_f + \sum_{e \in \text{edges}} \Gamma_e \delta l_e = \sum_a F_a \cdot \delta X_a \, , \tag{1}$$

where to get to the last expression one needs to express $\delta V_c$, $\delta A_f$ and $\delta l_e$ in terms of a virtual displacement of the vertices $\delta X_a$. Two and three-dimensional versions of the vertex model have been employed for many applications, for instance to study cell packing [23, 25], cell sorting [26], wound closure [27], cyst formation [28], tumourigenesis in tubular epithelia [29] among many other [19]. However because of their definition, vertex models do not explicitly resolve cortical flows on the cell surface. The effect of cell-cell adhesion is implicitly introduced in the surface tension $t_f$, which mixes together physical processes arising from cell-cell adhesion and surface forces in the cell membrane and in the actomyosin cortex. In vertex models with vertices positions as degrees of freedom, topological transitions leading to cell-neighbour exchange are encoded explicitly by formulating rules to change edges in the network.

At the single cell scale, a number of studies have shown the relevance of coarse-grained, continuum models to describe the mechanics of the cell surface. In animal cells, the cell surface is composed of a lipid membrane surrounding the actin cortex, a layer of cross-linked actin filaments undergoing continuous turnover of its constituents. Because of this turnover, the actin cortex behaves as a viscoelastic material with a characteristic remodelling time of tens of seconds [30, 31]. When looking at dynamics at time-scales beyond this relaxation time, the cortex can be seen as a viscous fluid layer. Because of the large cortical surface tension and effective 2D viscosity compared to the cell membrane [30, 32], models of cell surface mechanics have been focusing on the mechanics of the cortex. In this approach, an active fluid theory taking into account cellular cortical flows, gradients of active cytoskeletal tension and their regulation, and orientation and filament alignment in the actin cortex, has proven successful to describe the mechanics of cell polarisation, cell motility or cell division [33–42]. From a computational perspective, there has been a growing attention to the simulation of the dynamics of fluid interfaces both with prescribed [43–46] and with time-evolving shape [41, 42, 47–53]. In addition, recent discrete deformable cell models have described cellular aggregates by representing cells by triangular meshes with viscoelastic edges [54–57].

However, to our knowledge no computational framework has attempted to provide a physical description of three-dimensional cellular aggregates taking into account explicitly the mechanics of a single cell surface described as an active fluid surface, as well as cell-cell adhesions.

Here we bridge this gap and introduce a new modelling and simulation framework, and a freely available code [58], for the mechanics of cell aggregates in three dimensions (Fig 1). We describe cells as interacting active surfaces [59]. The governing equations for the cell surface

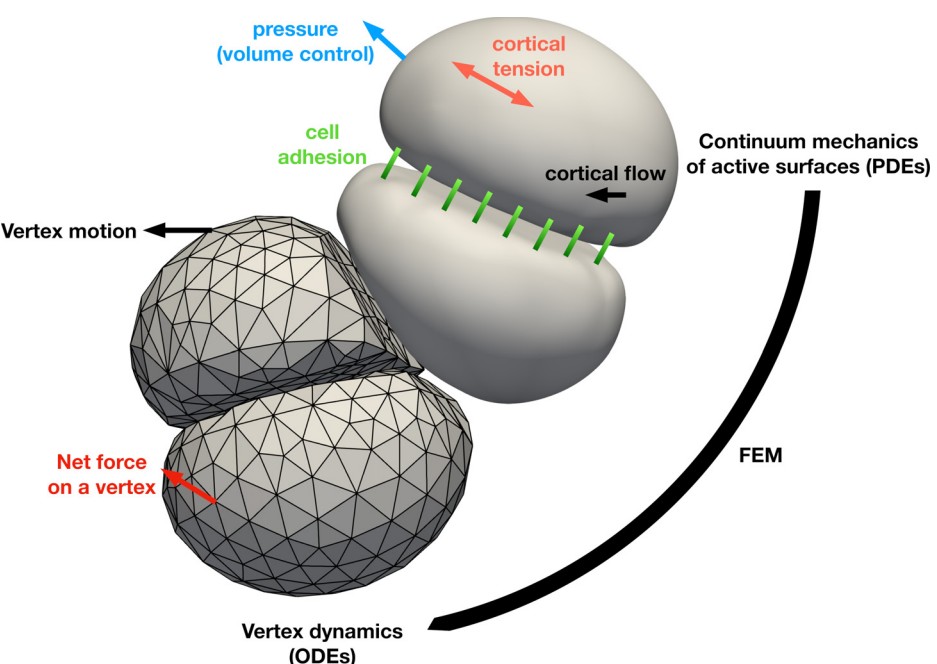

**Fig 1. Schematic of the interacting active surface framework.** A tissue is described by a collection of triangular meshes representing each cell. The dynamics of the tissue is described by the dynamics of the vertices making the cellular meshes, similar to how the movement of the vertices of a vertex model describe the deformation of a tissue. The motion of the cell mesh is obtained by coarse-graining continuum mechanics equations of a theory of active surfaces via the finite element method. In this theory, cortical flows, cortical tensions, intracellular pressures, bending moments and forces arising from cell-cell interactions are taken into account.

mechanics are discretised using a finite element method. In this method, each cell is represented by a three-dimensional mesh with vertices positions $X_a$. In analogy with Eq (1) in vertex models, we start from the virtual work theorem for interfaces (Eq (2)) and find the net forces at the vertices $F_a$ (Eq (11)), which vanish in the absence of inertia. This condition allows us to obtain cortical flows and cell shape changes. In this framework, cell-neighbour exchange appears as a natural output of the remodelling of cell-cell interactions and is not treated explicitly.

The main benefits of our method are that it (1) can incorporate complex descriptions of the physics of the cell surface, including sources of tension, in-plane and normal moments [59], (2) accounts for cell-cell adhesion explicitly through constitutive laws that can be adapted to represent different biological scenarios, (3) resolves cell shape and cortical flows with arbitrary resolution given by the mesh size of the finite element discretisation with a well-defined continuum limit.

The nonlinear equations of the model are treated computationally with the use of nonlinear solvers. Since we consider only the discretisation of the cell surface, the number of degrees of freedom is considerably smaller than in 3D Cellular Potts or phase-field models, both of which require a 3D discretisation. On the other hand, to capture cell shape, cortical flows and cell-cell interactions accurately, we need to use many more vertices per cell than in a vertex model, which leads to a greater computational cost. To limit computational time, the code is parallelised to allow each cell to be stored on a different partition, each possibly using a pool of cores (hybrid MPI-OpenMP method). As such, it is possible to simulate several tens of cells on a computing cluster.

We now turn to the description of our framework. In Section 2 we describe the mathematical formulation and the discretisation of our method. We show some examples of application in Section 3 and end in Section 4 with conclusions, summary and ideas for future work.

## 2 Materials and methods

In this section we describe the main elements of our framework. We start by introducing the governing equations for a single, isolated cell described as a fluid active surface. We use a virtual work formulation for the mechanics of a surface, together with a finite element discretisation, to obtain the equations that dictate the movement of the vertices of the mesh. We introduce cell-cell interactions, represented by an interaction potential between pairs of surfaces, that result in a force density and a tension acting on each cell. Finally, we describe a mesh reparametrisation method that allows simulations to handle large tangential deformations of the surface which can arise for continuously flowing fluid interfaces such as the cell cortex.

### 2.1 Governing equations and discretisation for a single surface

**2.1.1 Virtual work for interfaces.** Our starting point is the statement of virtual work for a closed interface $\mathcal{S}$ representing a single cell. We describe $\mathcal{S}$ with a time-dependent Lagrangian parametrisation $X(s^1, s^2)$ where $s^1$, $s^2$ are two surface coordinates and $X$ a point in the 3D space in which the surface is embedded; here and in the following, we ommit the dependency of $X$ on time for notational simplicity. Since we use a Lagrangian parametrisation, for fixed surface coordinates $s_*^1, s_*^2, X(s_*^1, s_*^2)$ follows the trajectory of a material particle. We denote the coordinates of the 3D cartesian basis by greek indices $\alpha, \beta, \ldots$, and the coordinates on the surface by latin indices $i, j. \ldots$. Here and elsewhere in the manuscript we use Einstein summation convention for repeated indices. Given the tangent vectors $e_i = \partial_i X$ one can compute the metric tensor $g_{ij} = e_i \cdot e_j$ and the curvature tensor $C_{ij} = -n \cdot \partial_i e_j$, where $n = (e_1 \times e_2)/|e_1 \times e_2|$ is the outer normal to the surface. Other notations of differential geometry are given in S1 Appendix 1. Here and in the following, we consider the limit of low Reynolds number where inertial terms are negligible, a limit relevant to the physics at the cell scale which is of interest here [60]. The mechanics of a single surface can then be described by the following statement of the principle of virtual work for $\mathcal{S}$ [59]:

$$\delta W = \int_{\mathcal{S}} dS \left\{ \frac{1}{2} \hat{t}^{ij} \delta g_{ij} + \bar{m}^{ij} \delta C_{ij} - f^\alpha \delta X^\alpha \right\} = 0 \ , \tag{2}$$

where $\delta X$ is an infinitesimal displacement of the surface, and $\delta g_{ij}$ and $\delta C_{ij}$ the associated infinitesimal variation of the metric and curvature tensors. Expressions for $\delta g_{ij}$ and $\delta C_{ij}$ in terms of $\delta X^\alpha$ are given in S1 Appendix 2. Eq (2) relates infinitesimal variations of geometric quantities of the interface to their work conjugates: the external force density $f^\alpha$ coupled to the surface displacement, the tension tensor $\hat{t}^{ij}$ related to metric variations, and the bending moment tensor $\bar{m}^{ij}$ coupled to variations of the curvature tensor. Eq (2) is equivalent to the statement of balance of linear and angular momentum at low Reynolds number (S1 Appendix 3). We have neglected external torques and internal normal moments for simplicity, which lead to extra terms in Eq (2) (S1 Appendix 3).

**2.1.2 Constitutive laws.** To describe the mechanics of a single surface, we need to specify constitutive equations for the mechanical tensors $\hat{t}^{ij}$ and $\bar{m}^{ij}$. We denote by $v = \partial_t X$ the velocity field on the cell surface. We assume that the cell surface can be represented as an active viscous layer, with a bending rigidity and a spontaneous curvature. As a result, we use the following

constitutive equations:

$$\hat{t}^{ij} = \hat{t}_{\mathrm{d}}^{ij} + \hat{t}_{\mathrm{a}}^{ij} + \hat{t}_{\mathrm{e}}^{ij} \, ,$$

$$\hat{t}_{\mathrm{d}}^{ij} = 2\eta v^{ij}, \quad \hat{t}_{\mathrm{a}}^{ij} = \gamma g^{ij}, \quad \hat{t}_{\mathrm{e}}^{ij} = \kappa \left( C_k^k - C_0 \right) \left[ \frac{1}{2} \left( C_k^k - C_0 \right) g^{ij} - 2C^{ij} \right] \, , \tag{3}$$

$$\bar{m}^{ij} = \bar{m}_{\mathrm{e}}^{ij} = \kappa ( C_k^k - C_0 ) g^{ij} \, ,$$

where $v_{ij}$ is the strain-rate tensor on the surface:

$$v_{ij} = \frac{1}{2} \left[ \nabla_i v_j + \nabla_j v_i + 2C_{ij} v_n \right] \, , \tag{4}$$

with $\nabla_i$ the covariant derivative operator defined in S1 Appendix 1. Here, $\eta$ is the surface viscosity; for simplicity we do not explicitly distinguish between the shear and bulk viscosity of the surface. $\gamma$ is the surface tension of the cell, which is not necessarily homogeneous on the cell surface. As we expect this contribution to largely arise from active forces generated in the actomyosin cortex [30], we later refer to it as the active tension. The contributions $\hat{t}_{\mathrm{e}}^{ij}$ and $\bar{m}_{\mathrm{e}}^{ij}$ arise from an effective Helfrich free energy penalising the membrane curvature with bending modulus $\kappa$ and spontaneous curvature $C_0$ (S1 Appendix 4).

The external force density acting on a single cell is split into two contributions

$$\boldsymbol{f} = \boldsymbol{f}_{\mathrm{e}} + \boldsymbol{f}_{\mathrm{d}} \, , \tag{5}$$

arising respectively from the action of the pressure difference across the surface

$$f_{\mathrm{e}}^{\alpha} = P n^{\alpha} \, , \tag{6}$$

and from an effective external friction force density

$$f_{\mathrm{d}}^{\alpha} = -\xi v^{\alpha} \, , \tag{7}$$

where $\xi$ is a friction coefficient. In the following, the pressure $P$ is adjusted to impose a value of the cell volume. Eq (2), together with the constitutive Eqs (3)–(7), leads to a complete set of equations to determine the velocity field $\boldsymbol{v}$ on the cell surface.

The constitutive laws Eqs (3)–(7) are a simple choice of physical description of the surface. Other terms, for instance additional viscous or active bending moments, could be introduced: linear irreversible thermodynamics provides with a set of additional possible terms that could play a role in the dynamics of isotropic, polar or nematic active surfaces and could be added to the framework described here [59, 61, 62]. Furthermore, by neglecting possible dependencies of the constitutive laws Eqs (3)–(7) on cortical density $\rho$, we have implicitly assumed that $\rho$ is homogeneous and constant; this is an appropriate approximation if cortical turnover is faster than the characteristic time for the relaxation dynamics of the cortex by flow and deformation. Furthermore, the active tension $\gamma$ could be a function of the concentration of molecular actuators, such as myosin molecular motors. We describe how cortical density or other concentration fields on the cell surface can be incorporated into our formulation in the Discussion (see Eq (42)).

**2.1.3 Finite element discretisation.** We now discretise the geometry of the surface using finite elements based on a triangular control mesh of $N_e$ triangles or elements and $N_v$ vertices (Fig 2). The parametrisation of $\mathcal{S}$ is of the form

$$\boldsymbol{X}(s^1, s^2) = \sum_a \boldsymbol{X}_a B_a(s^1, s^2) \, , \tag{8}$$

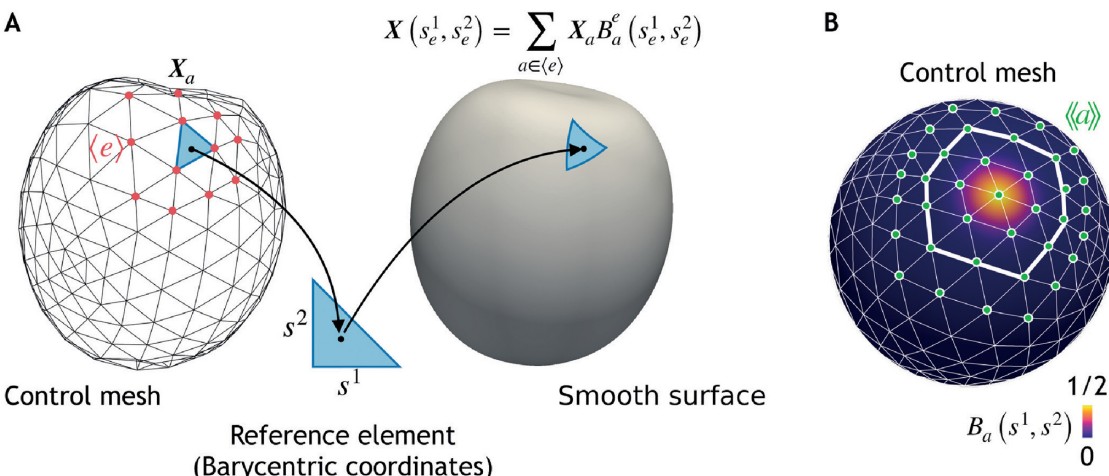

**Fig 2. A smooth surface $\mathcal{S}$ representing a cell is obtained based on a triangular control mesh with vertex positions $X_a$ (A) and a set of basis functions per vertex $a$ (B).** (A) To define the mapping between the control mesh and the cell surface, the barycentric coordinates of points in a triangular element $e$ in the control mesh $s_e^1, s_e^2$, which span a reference triangle, are used to define a point on the cell surface $\mathcal{S}$, $X(s_e^1, s_e^2)$ (Eq (9)). Points $X(s_e^1, s_e^2)$ are obtained by summing basis functions $B_a(s_e^1, s_e^2)$, weighted by $X_a$, over vertices $a$ whose basis functions have a non-zero contribution to this element, an ensemble denoted $\langle e \rangle$. (B) Example of the basis function associated to a vertex in the mesh. For Loop subdivision surfaces basis functions, the basis function spans the first and second rows of elements surrounding the vertex (thicker white line). The vertices that interact with vertex $a$ in the same cell, represented by the set $\langle\langle a \rangle\rangle$ (green) are formed by the first, second, and third nearest neighbours in the mesh.

where the sum is taken over vertices of the control mesh, labelled by $a$, $X_a$ is the position of vertex $a$ on the control mesh, and $B_a(s^1, s^2)$ is its basis function, defined in terms of coordinates ($s^1$, $s^2$) on the control mesh. Given that $\mathcal{S}$ may have bending or in-plane moments coupled to variations of the curvature tensor and the Christoffel symbols in the differential virtual work (see Eq (2) and Eq. 34 in S1 Appendix 3), the basis functions $B_a$ must be chosen to have second order derivatives that are square-integrable; here we follow [41, 63, 64] and use Loop subdivision surfaces, which lead to smooth surfaces that satisfy this condition by construction. We note that, in practice, a consistent parametrisation ($s^1, s^2$) of the entire control mesh is not needed: indeed one can consider the surface $\mathcal{S}$ as a union of surface elements associated to each triangle $e$ of the control mesh, which are given in terms of barycentric coordinates $s_e^1, s_e^2$ of the element $e$ by

$$X(s_e^1, s_e^2) = \sum_{a \in \langle e \rangle} X_a B_a^e(s_e^1, s_e^2) \ , \tag{9}$$

see Fig 2. Here we have denoted by $\langle e \rangle$ the set of vertices whose basis functions are nonzero in element $e$, and $B_a^e(s_e^1, s_e^2)$ corresponds to the contribution of the basis function $B_a$, within element $e$, parametrised by the barycentric coordinates of $e$. For Loop subdivision surfaces, $\langle e \rangle$ is formed by the vertices in the triangular element $e$, and all first neighbours of these vertices.

The discretisation of the surface (Eq (8)) transforms the virtual work principle (Eq (2)) into an expression of the form

$$\delta W = \sum_a F_a\left(\left\{X_b, \dot{X}_b\right\}_{b \in \langle\langle a \rangle\rangle}\right) \cdot \delta X_a = 0 \ . \tag{10}$$

Here $F_a$, which can be interpreted as the net force on vertex $a$, can be obtained by substituting, in the differential virtual work, the analytical expressions for the variations $\delta g_{ij}$ and $\delta C_{ij}$ in terms of $\delta X = \sum_a B_a(s^1, s^2)\delta X_a$ (S1 Appendix 6). We have denoted by $\langle\langle a \rangle\rangle$ the set of vertices interacting with vertex $a$, which for Loop subdivision surfaces is formed by its first, second and

third ring of neighbours (Fig 2B). Since Eq (10) has to be satisfied for any $\delta X_a$, the net forces on the vertices need to vanish

$$\boldsymbol{F}_a\left(\left\{\boldsymbol{X}_b, \dot{\boldsymbol{X}}_b\right\}_{b\in\langle\langle a\rangle\rangle}\right) = 0 \ . \tag{11}$$

Through this discretisation, we transform the original continuum problem into a set of coupled ordinary differential equations (ODEs). These ODEs need to be discretised in time to be resolved computationally; in the following we denote by ($n$) the $n$–th time step of the time evolution. Here we employ a semi-implicit Euler discretisation, where terms arising from the effective bending energy and active tension are discretised in a fully implicit manner, whereas viscous and frictional terms are treated explicitly. This particular choice leads to a variational time-integrator that preserves the dissipative structure of the dynamics for a homogeneous and time-independent active tension [41]. This implies that the tension and bending moment tensors at step $n$ are evaluated as:

$$
\begin{aligned}
\hat{t}_{ij}^{(n)} &= \eta \frac{g_{ij}^{(n+1)} - g_{ij}^{(n)}}{\Delta t^{(n)}} + \gamma g_{ij}^{(n+1)} \\
&\quad + \kappa\left(C^{(n+1)k}{}_k - C_0\right)\left[\frac{1}{2}\left(C^{(n+1)k}{}_k - C_0\right)g_{ij}^{(n+1)} - 2C_{ij}^{(n+1)}\right] ,
\end{aligned}
\tag{12}
$$

$$\bar{m}_{ij}^{(n)} = \kappa\left(C^{(n+1)k}{}_k - C_0\right)g_{ij}^{(n+1)} ,$$

which can be written in terms of $\boldsymbol{X}_a^{(n+1)}$ and $\boldsymbol{X}_a^{(n)}$ through the relation between the surface definition and the position of vertices, Eq (8). To discretise the strain rate tensor, we have used its relation with the rate of change of components of the metric tensor (see S1 Appendix 2). Plugging these expressions in the differential virtual work Eq (2) allows us to transform Eq (11) into a set of (nonlinear) algebraic equations (S1 Appendix 6)

$$\boldsymbol{F}_a\left(\left\{\boldsymbol{X}_b^{(n)}, \boldsymbol{X}_b^{(n+1)}\right\}_{b\in\langle\langle a\rangle\rangle}, P^{(n+1)}\right) = 0 \ , \tag{13}$$

which can be solved using a Newton-Raphson method together with the discretisation of the volume constraint, which is imposed through the nonlinear constraint (S1 Appendix 6.2):

$$F_P\left(\boldsymbol{X}_a^{(n+1)}\right) = V_0 \ . \tag{14}$$

Here $P^{(n+1)}$ is the intracellular pressure, playing the role of a Lagrange multiplier imposing conservation of volume (S1 Appendix 6.2). In this method, the solution is updated by $\boldsymbol{X}_a^{(n+1)} \leftarrow \boldsymbol{X}_a^{(n+1)} + \Delta\boldsymbol{X}_a$, $P^{(n+1)} \leftarrow P^{(n+1)} + \Delta P$ where $\Delta\boldsymbol{X}_a$ and $\Delta P$ satisfy the linearised equations

$$
\begin{aligned}
\frac{\partial\boldsymbol{F}_a}{\partial\boldsymbol{X}_b^{(n+1)}}\Delta\boldsymbol{X}_b &= -\boldsymbol{F}_a \ , \\
\frac{\partial\boldsymbol{F}_a}{\partial P^{(n+1)}}\Delta P &= -\boldsymbol{F}_a \\
-\frac{\partial F_P}{\partial\boldsymbol{X}_a^{(n+1)}}\Delta\boldsymbol{X}_a &= F_P - V_0 \ ,
\end{aligned}
\tag{15}
$$

until the norm of $\boldsymbol{F}_a$ and $F_P - V_0$ is below a given tolerance. Here $\partial\boldsymbol{F}_a/\partial\boldsymbol{X}_b^{(n+1)}$, $\partial\boldsymbol{F}_a/\partial P^{(n+1)}$, and $-\partial F_P/\partial\boldsymbol{X}_a^{(n+1)}$ form the tangent matrix. Because of the way we perform the discretisation,

the tangent matrix is symmetric, in particular $-\partial F_P / \partial X_a^{(n+1)} = \partial \boldsymbol{F}_a / \partial P^{(n+1)}$, see S1 Appendix (71)-(74), (83) and (84). Because only vertices in $\langle\langle a \rangle\rangle$ interact with $a$, this matrix is sparse and the linear system can be solved efficiently with an iterative solver. Our computational framework makes use of Trilinos [65] to handle all linear algebra objects, including sparse matrices, in parallel. To solve Eq (15), we employ a GMRES solver from the Belos package of Trilinos.

## 2.2 Forces arising from cell-cell interactions

Cell-cell adhesion modifies the previous equations by introducing interactions between the vertices of interacting surfaces. We now consider a set of surfaces $\mathcal{S}_I$, $I = 1\ldots N$ describing $N$ interacting cells. We assume that a pair of distinct cells $I, J$ interact via an effective energy:

$$\mathcal{F}_{IJ}[\boldsymbol{X}_I, \boldsymbol{X}_J] = \int_{\mathcal{S}_I} dS_I \int_{\mathcal{S}_J} dS_J \varphi(|\boldsymbol{X}_I - \boldsymbol{X}_J|) \ , \tag{16}$$

where $\varphi$ is an effective adhesion potential. We first give a microscopic motivation of $\varphi$ in Section 2.2.1 and then discuss how Eq (16) can be used to derive the governing equations of a cell aggregate in Section 2.2.2.

**2.2.1 Microscopic motivation.** The form of the effective energy (Eq (16)) can be motivated microscopically by considering an ensemble of stretchable linkers connecting pairs of surfaces, which quickly equilibrate by binding and unbinding to cell surfaces, and whose free concentration is set by contact with a reservoir (Fig 3A). We characterise such an ensemble by a two-point concentration field $c_{IJ}(\boldsymbol{X}_I, \boldsymbol{X}_J)$, which quantifies the number of bound linkers joining the points $\boldsymbol{X}_I$ and $\boldsymbol{X}_J$ per unit area of the first and second surfaces (thus, it has units of the inverse of an area squared). The concentration $c_I$ denotes the concentration of unbound, free linkers in cell $I$, with units of the inverse of an area. In the dilute limit, the free energy of this ensemble can be written as:

$$\begin{aligned} \mathcal{F}^{\mathrm{micro}} = \quad & \sum_I \int_{\mathcal{S}_I} dS_I k_{\mathrm{B}} T c_I \left( \log \frac{c_I}{c_0} - 1 \right) \\ & + \sum_{\langle I,J \rangle} \int_{\mathcal{S}_I} dS_I \int_{\mathcal{S}_J} dS_J c_{IJ} \left[ k_{\mathrm{B}} T \left( \log \frac{c_{IJ}}{c_0^2} - 1 \right) + \phi\left(|\boldsymbol{X}_I - \boldsymbol{X}_J|\right) \right] \ , \end{aligned} \tag{17}$$

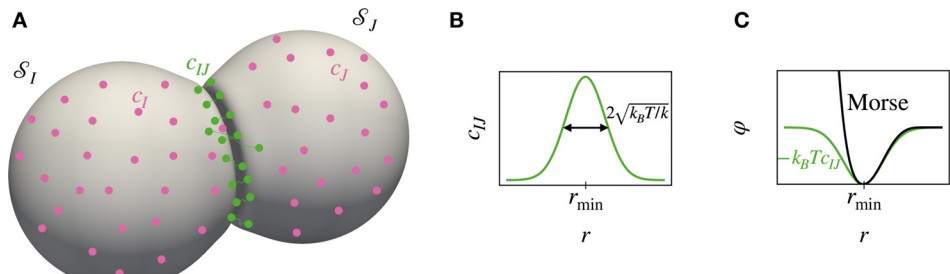

**Fig 3. (A) Physical picture of two interacting cells, *I* and *J*, that adhere via stretchable linkers.** The concentrations $c_I$ and $c_J$ describe the number of free linkers in cells *I* and *J* per unit area. When free linkers from cells *I* and *J* are in close proximity, they can react to generate a linker joining the two cells. The ensemble of connected linkers is characterised by a two-point concentration field $c_{IJ}(\boldsymbol{X}_I, \boldsymbol{X}_J)$ with units of inverse of an area squared. (B) Concentration of connected linkers as a function of the distance $r = |\boldsymbol{X}_I - \boldsymbol{X}_J|$, for linear elastic bonds with stiffness $k$ and preferred elongation $r_{\mathrm{min}}$, fast equilibration of linker density and assuming that free linkers are in contact with a reservoir imposing their equilibrium concentration. (C) Green: effective potential of interaction of two points on two surfaces as a function of their distance $r$. Black: the Morse potential resembles the effective potential of interaction but with additional short-range repulsion.

where $k_B$ is the Boltzmann constant and $T$ the temperature. Here and in the following, we denote distinct cell pairs $\langle I, J \rangle$, such that in sums taken over $\langle I, J \rangle$, each pair is counted only once. The first sum in Eq 17 corresponds to the free energy of free linkers, described as an ideal solution in contact with a reservoir imposing a chemical potential. This chemical potential determines the value of $c_0$. The second sum corresponds to the free energy of bound linkers, described as an ideal solution, with an energy per linker dependent on the linker elongation, quantified by the potential $\phi$. The potential is defined such that the force sustained by a linker of length $r$ is $-\phi'(r)$. One can then show that if linkers quickly relax to their Boltzmann distribution with respect to surfaces $I, J$ with a fixed shape and free linkers are in contact with a reservoir imposing their equilibrium distribution, then $c_{IJ} = c_0^2 \exp\left[-\frac{\phi(|\boldsymbol{X}_I - \boldsymbol{X}_J|)}{k_B T}\right]$. With this assumption, the free energy of interaction of two surfaces $I, J$ is (S1 Appendix 5)

$$\mathcal{F}_{IJ} = -k_B T c_0^2 \int_{\mathcal{S}_I} dS_I \int_{\mathcal{S}_J} dS_J \exp\left[-\frac{\phi(|\boldsymbol{X}_I - \boldsymbol{X}_J|)}{k_B T}\right] , \qquad (18)$$

which gives a relation between the microscopic behaviour of the linkers and the potential $\varphi$ introduced in Eq (16). For the particular choice of $\phi(r) = \phi_0 + k(r - r_{min})^2/2$ with $k$ the bond stiffness and $r_{min}$ its reference length, one obtains:

$$\varphi(r) = -k_B T c_0^2 \exp\left(-\frac{\phi_0}{k_B T}\right) \exp\left[-\frac{k(r - r_{min})^2}{2k_B T}\right] , \qquad (19)$$

which corresponds to an inverted Gaussian with centre at the equilibrium length $r_{min}$, width $\sqrt{k_B T/k}$ and depth $D = k_B T c_0^2 \exp\left(-\phi_0/(k_B T)\right)$ (Fig 3B and 3C).

However, this description still does not take into account short range repulsion between two cell surfaces. This could be taken into account by introducing a second repulsive interaction potential between surfaces. Here we choose instead to introduce a convenient effective potential of interaction, the Morse potential

$$\varphi_{Morse}(r) = D\left\{\left[1 - \exp\left(\frac{r_{min} - r}{l}\right)\right]^2 - 1\right\} , \qquad (20)$$

which like the interaction potential in Eq (18), vanishes for $r \to \infty$, has a minimum at $r = r_{min}$ with minimum value $-D$, and is also peaked around its minimum with characteristic length $l$. To match the second-order expansion of Eq (19) around $r_{min}$, one would choose $l = \sqrt{2k_B T/k}$ (Fig 3C). In addition, for $l \ll r_{min}$ the Morse potential exhibits a sharp short-range repulsion.

Although $\varphi_{Morse}(r)$ decays with $r$ rapidly, it is convenient to have a strict cut-off on its range, to limit interacting vertices of the meshes. Therefore, we further multiply $\varphi_{Morse}(r)$ by a smooth step function:

$$\varphi(r) = \varphi_{Morse}(r) w(r; r_{min}, r_{min} + 3l) . \qquad (21)$$

where

$$w(r; r_1, r_2) = \begin{cases} 1 & \text{if } r \leq r_1 , \\ \left(\frac{r_2 - r}{r_2 - r_1}\right)^3\left[6\left(\frac{r_2 - r}{r_2 - r_1}\right)^2 - 15\left(\frac{r_2 - r}{r_2 - r_1}\right) + 10\right] & \text{if } r_1 < r < r_2 , \\ 0 & \text{if } r \geq r_2 . \end{cases} \qquad (22)$$

This ensures that the potential $\varphi(r)$ goes to zero exactly at a distance $r_{min} + 3l$ with first and

second order continuous derivatives; a convenient property to solve numerically the non-linear equations (Eq (27)).

**2.2.2 Governing equations of a cell aggregate.** The virtual work differential of the cell aggregate can then be written as:

$$\delta W = \sum_I \delta W_I + \sum_{\text{cell pairs} \langle I,J \rangle} \delta F_{IJ} \;, \tag{23}$$

where the contribution $\delta W_I$ is given by Eq (2) with infinitesimal displacement ($\delta X_I$), metric variation ($\delta g_{Iij}$) and curvature variation ($\delta C_{Iij}$) of cell $I$, and with the tension tensor $\hat{t}_{ij}$, bending moment tensor $\bar{m}_{ij}$, and external force density excluding cell-cell interaction forces $f$, given by the constitutive Eqs (3)–(7).

Variations of the interaction free energy lead to

$$\begin{aligned} \delta \mathcal{F}_{IJ} \;\; = \int_{\mathcal{S}_I} dS_I \int_{\mathcal{S}_J} dS_J \; & \left\{ \varphi'\left(|\boldsymbol{X}_I - \boldsymbol{X}_J|\right) \frac{X_I^\alpha - X_J^\alpha}{|\boldsymbol{X}_I - \boldsymbol{X}_J|} (\delta X_I^\alpha - \delta X_J^\alpha) \right. \\ & \left. + \frac{1}{2} \varphi\left(|\boldsymbol{X}_I - \boldsymbol{X}_J|\right) \left(g_I^{ij} \delta g_{Iij} + g_J^{ij} \delta g_{Jij}\right) \right\} \;. \end{aligned} \tag{24}$$

Comparing this expression with Eq (2), each cell interaction with a cell $J$ is contributing an additional external force density on cell $I$, $f_{IJ}$, and an additional isotropic tension to cell $I$, $\hat{t}_{IJ}^{ij}$:

$$f_{IJ} = -\int_{\mathcal{S}_J} dS_J \; \varphi'\left(|\boldsymbol{X}_I - \boldsymbol{X}_J|\right) \frac{\boldsymbol{X}_I - \boldsymbol{X}_J}{|\boldsymbol{X}_I - \boldsymbol{X}_J|} \;, \tag{25}$$

$$\hat{t}_{IJ}^{ij} = \int_{\mathcal{S}_J} dS_J \; \varphi(|\boldsymbol{X}_I - \boldsymbol{X}_J|) g_I^{ij} \;. \tag{26}$$

In addition, although this point is not directly apparent from Eq (24), the variation $\delta \mathcal{F}_{IJ}$ can be written only in terms of normal displacements, provided that $\varphi$ only depends on $|\boldsymbol{X}_I - \boldsymbol{X}_J|$ (Eq. (53) in S1 Appendix 3). This shows that the net driving force from the interaction potential, including the effect of both the force density $f_{IJ}$ and tension $\hat{t}_{IJ}^{ij}$, has a vanishing tangential component. Intuitively, the interaction energy does not change if cells do not change shape, so it cannot generate a driving force for tangential motion.

Following the same procedure as in the previous section, the condition $\delta W = 0$, with $\delta W$ defined in Eq (23), gives rise to a set of (nonlinear) algebraic equations of the form

$$\boldsymbol{F}_{I,a}\left( \left\{ \boldsymbol{X}_{J,b}^{(n)}, \boldsymbol{X}_{J,b}^{(n+1)} \right\}_{J,b \in \langle\langle I,a \rangle\rangle}, P_I^{(n+1)} \right) = 0 \;. \tag{27}$$

Here $\langle\langle I, a \rangle\rangle$ identifies the set of vertices (identified as the pair of labels $J$ for the cell considered and $b$ for the vertex considered) that interact with vertex $a$ in cell $I$.

## 2.3 Surface reparametrisation

The method discussed in the previous sections is based on a Lagrangian scheme, such that a node of the mesh flows with the material particles of the interface. This, however, can lead to large in-plane distortions, notably if surface tension gradients generate in-plane flows [41, 50]. To compensate for the resulting mesh distortion, we describe here a reparametrisation method, in the spirit of Refs. [63, 64]. In this method, vertices of the control mesh move tangentially to the surface to minimise an effective mesh quality energy. We stress that this step

does not bear any physical meaning. Here we discuss again a single cell. Given the mesh of a cell, we define the energy

$$\mathcal{F}_{\text{mesh}}\left(\left\{\boldsymbol{X}_a\right\}_{a=1}^N\right) = \sum_e f(I_e, J_e) \;,\tag{28}$$

where the sum is performed over the triangles of the control mesh, and

$$I_e = \frac{\sqrt{3}}{2} \cdot \frac{l_{e1}^2 + l_{e2}^2 + l_{e3}^2}{A_e} \;, \quad J_e = \sqrt{\frac{A_e}{\langle A \rangle}} \;,\tag{29}$$

where $A_e$ is the area of the triangle $e$ and $l_{e1}, l_{e2}, l_{e3}$ its side lengths. $I_e$ and $J_e$ represent the invariants (trace and square root of the determinant) of the Cauchy-Green deformation tensor [66] assuming a reference equilateral triangle of size $\langle A \rangle = \Sigma_e A_e/N_e$, where the sum is taken over the $N_e$ triangles of a meshed surface. To specify the free energy $f$, we use the Neohookean energy $f(I_e, J_e) = \mu I_e + \lambda(J_e - 1)^2$ [66]. Note that we define this energy on the mesh rather than on $\mathcal{S}$. We want to minimise Eq (28), but with the restriction that the cell shape $\mathcal{S}$ does not change, so that this operation corresponds to a surface reparametrisation without surface deformation. For this, we evolve the position of the vertices of the mesh according to velocities $\dot{\boldsymbol{X}}_a$. These velocities are obtained by introducing a continuous velocity field on the surface $\boldsymbol{v}(s^1, s^2) = \sum_a B_a(s^1, s^2)\dot{\boldsymbol{X}}_a$, and by solving the following equations for the vertices velocities:

$$\frac{\partial \mathcal{F}_{\text{mesh}}}{\partial X_a^\alpha} + \int_{\mathcal{S}} [\xi_m v^\alpha + p n^\alpha] B_a dS = 0 \;,\tag{30}$$

$$\int_{\mathcal{S}} v^\alpha n^\alpha B_a dS = 0 \;,\tag{31}$$

where $\xi_m$ is a fictitious mesh friction coefficient, $p(s^1, s^2) = \Sigma_a p_a B_a(s^1, s^2)$ plays the role of a normal pressure, here a field enforcing the condition that the normal flow vanishes in a weak sense, Eq (31). This leads to the relaxation of the free energy $\mathcal{F}_{\text{mesh}}$ following a gradient-flow, with vertices constrained to the shape of $\mathcal{S}$, where the constraint is enforced weakly, i.e. in a finite element sense (S1 Appendix 7, S1 Video).

## 3 Results

We now discuss applications of the interacting active surface framework. We first consider flows in a single spherical cell driven by gradient of cortical tension, a set-up which allows to compare simulation results to an analytical solution. We then examine the shape of the simplest multicellular aggregate, a doublet formed by two cells, when the two participating cells have equal or different tensions. Next, we consider an aggregate of cells assembled in a planar configuration, recapitulating the organisation of a small epithelial island. Finally, we introduce cell divisions in our framework and simulate the growth of a three-dimensional cell aggregate from a single cell. In the following we introduce a reference length scale $\ell = (3V^*/(4\pi))^{\frac{1}{3}}$ which corresponds to cell radius of a spherical cell, a reference surface tension $\bar{\gamma}$, and a reference time scale $\tau = \eta/\bar{\gamma}$, which corresponds to the characteristic time scale of cortical flows. We use these reference quantities for normalisation of other quantities.

### 3.1 Single cell: Convergence with mesh size

We first consider flows driven by gradients of active tension in a single cell. This allows us to test the convergence of the numerical method for the dynamics of a single spherical cell, since

we can compare the velocity field resulting from the method discretisation to an analytical solution using spherical harmonics; we note here that we only consider the velocity field in the initial, undeformed sphere where the solution with spherical harmonics is exact. We consider a pattern of surface tension on a spherical surface of radius $\ell$, given by

$$\gamma = \sum_{A=0}^{\infty}\sum_{a=-A}^{A} \gamma^{Aa} Y^{Aa} \ , \tag{32}$$

where $Y^{Aa}$ is the spherical harmonic of degree $A$ and order $a$. The volume enclosed by the surface is assumed to be subjected to a uniform pressure difference $P$. The resulting velocity field can then be written as (S1 Appendix 8, [67, 68])

$$\boldsymbol{v} = (\partial_i \phi)\, \boldsymbol{e}^i + v_n \boldsymbol{n} \ , \tag{33}$$

where the functions $\phi$ and $v_n$ can also be expanded in spherical harmonics, $\phi = \sum_{A=0}^{\infty}\sum_{a=-A}^{A} \phi^{Aa} Y^{Aa}$, $v_n = \sum_{A=0}^{\infty}\sum_{a=-A}^{A} v_n^{Aa} Y^{Aa}$, and the coefficients read:

$$\phi^{Aa} = \frac{\gamma^{Aa}\dfrac{\xi\ell^4}{4\eta^2}}{A(A+1)\left(1+\dfrac{\xi\ell^2}{2\eta}\right)+\left(2+\dfrac{\xi\ell^2}{2\eta}\right)\left(\dfrac{\xi\ell^2}{2\eta}-1\right)} \ , \quad A > 0 \ , \tag{34}$$

$$v_n^{Aa} = \frac{1}{\left(2+\dfrac{\xi\ell^2}{2\eta}\right)\ell}\left[A(A+1)\phi^{Aa} - \frac{\ell^2}{\eta}\gamma^{Aa} + \frac{\sqrt{4\pi}\ell^3}{2\eta}\delta^{A0}(P\ell+\kappa(2/\ell-C_0)C_0)\right] \ . \tag{35}$$

In Fig 4, we consider flows resulting from a surface tension profile given by the coefficients $\gamma^{00} = 2\sqrt{\pi}\bar{\gamma}\ell$, $\gamma^{Aa} = \bar{\gamma}\ell a^2/A^2$ if $1 < A \leq 4$ and $\gamma^{Aa} = 0$ for $A > 4$, and from the imposed inner pressure $P = 2\bar{\gamma}/\ell$ (Fig 4A). We obtain the velocity field analytically ($\boldsymbol{v}^*$, Fig 4B) and numerically ($\boldsymbol{v}$, Fig 4C), for different mesh sizes. We then compute the $L_2$ norm of the error $\boldsymbol{v} - \boldsymbol{v}^*$, i.e. $L_2(|\boldsymbol{v}-\boldsymbol{v}^*|) = \sqrt{\int_{\mathcal{S}}|\boldsymbol{v}-\boldsymbol{v}^*|^2 dS}$, with respect to the $L_2$ norm of $\boldsymbol{v}^*$, $L_2(|\boldsymbol{v}^*|) = \sqrt{\int_{\mathcal{S}}|\boldsymbol{v}^*|^2 dS}$.

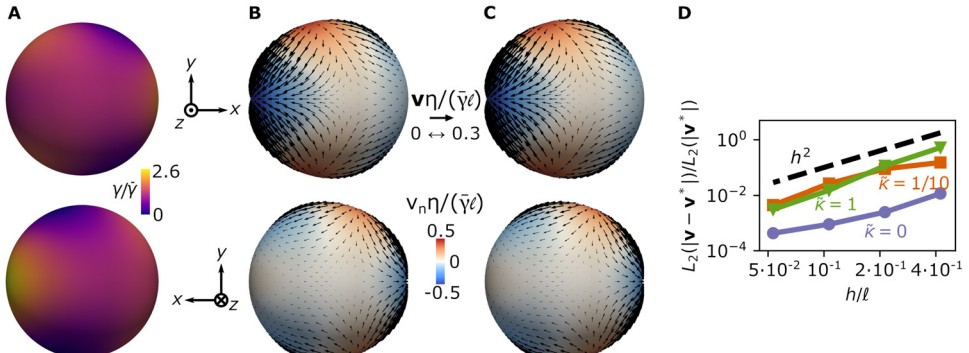

**Fig 4. Single cell dynamics and convergence of the numerical framework.** (A) An inhomogeneous pattern of active surface tension $\gamma$ is imposed on a spherical surface. (B) Analytically computed velocity field generated by the active tension profile in A. The velocity field is decomposed into its normal (colormap) and tangential (arrows) components. (C) Numerical solution for the velocity profile generated by the active tension profile in A. (D) Discretisation error, evaluated here in terms of the $L_2$ norm of the difference between the analytical and numerical solutions for the velocity, as a function of the average mesh size $h$ and for $\tilde{\kappa} = \kappa/(\bar{\gamma}\ell^2) = 0$ (blue), $\tilde{\kappa} = 0.1$ (orange) and $\tilde{\kappa} = 1$ (green). Other parameters are $\xi\ell^2/\eta = 4$, $C_0\ell = 1$.

We find an excellent agreement between the exact and numerically obtained velocity field (Fig 4B–4D). The corresponding error scales with $(h/\ell)^2$, where $h$ is the average mesh size, in line with the reported convergence rate of subdivision surfaces for other systems of partial differential equations [69] (Fig 4D).

## 3.2 Shape and dynamics of an adhering cell doublet

We now discuss the equilibrium shape of an adhering cell doublet. In this and the following sections, the cell pressure difference $P_I$ is imposed as a Lagrange multiplier enforcing the condition $V_I = V_0$, with $V_I$ the cell volume and $V_0 = \frac{4}{3}\pi\ell^3$ a reference volume, and we assume that $C_0 = 0$. With these choices, 5 normalised, non-dimensional parameters have to be specified for each simulation: $\tilde{\xi} = \xi\ell^2/\eta, \tilde{l} = l/\ell, \tilde{r}_{\min} = r_{\min}/\ell, \tilde{D} = Dr_{\min}l/\bar{\gamma}$, and $\tilde{\kappa} = \kappa/(\bar{\gamma}\ell^2)$.

In the following we set $\tilde{\xi} = 10^{-3}$ so that the effect of friction is small. A cadherin bond has a typical length $\sim$ 15–30 nm [70] and a typical actomyosin cortex thickness is 200nm [71], both much smaller than the typical radius of a cell, $\sim$ from a few to tens of μm. Therefore, in the interaction of potential of cell surface we take $\tilde{r}_{\min}, \tilde{l} \ll 1$. For simplicity, in the following we constrain $\tilde{r}_{\min} = 3\tilde{l}$. We typically choose values between $\tilde{r}_{\min} = 0.06, \tilde{l} = 0.02$ and $\tilde{r}_{\min} = 0.12, \tilde{l} = 0.04$, which for a cell radius of 5μm, correspond to $r_{\min} = 300 - 600$nm and $l = 100 - 200$nm.

We note that if the reference cell volume $V_0$ is constant, the system minimises the net effective free energy $\mathcal{F} = \sum_I \int_{\mathcal{S}_I} (\bar{\gamma} + \kappa(C_i{}^i)^2/2)dS^I + \sum_{\langle I,J\rangle} \mathcal{F}_{IJ}$, subjected to the constraint $V_I = V_0$, where the first term represents an effective energy for the active tension $\bar{\gamma}$ and $\mathcal{F}_{IJ}$ is defined in Eq (16). We thus expect the system to eventually reach an equilibrium state with vanishing cortical flows.

We first analyse the behaviour of a doublet of identical cells. We initialise the simulation by putting two spherical cells close to each other, such that they are within the interaction range of the potential $\varphi(r)$ (Eq (21)) without touching. As expected, after an initial transient and contact growth, the doublet reaches an equilibrium shape (Fig 5A and 5B). Increasing the relative adhesion strength $\tilde{D}$ leads to an increasing adhesion patch and a lower cell pressure (Fig 5C and 5D and S1B Fig). The value of the normalised distance $\tilde{l}$ modulates the distance between the two cells (Fig 5E). For $\tilde{D}$ beyond a threshold which depends on $\tilde{r}_{\min}$ and $\tilde{l}$, the adhesion patch develops a buckling instability. We found that this instability eventually leads to self-intersection of the computational mesh, as there is no energy contribution in our framework preventing such self-intersection (S1A Fig). Varying the bending modulus $\tilde{\kappa}$, but maintaining relatively small values ($\tilde{\kappa} \leq 10^{-2}$), we find slightly smoother shapes at the boundary of the adhesion patch as well as a slight modulation of the threshold for buckling instability of the contact zone, with larger $\tilde{\kappa}$ leading to higher stability (Fig 5C and 5E).

To interpret these results, we turn to an approximate analysis of the shape of a doublet. Assuming a small bending rigidity $\tilde{\kappa} \ll 1$, we approximate the cell doublet by two spherical caps of height $h_c$ and base of radius $r_c$, forming an adhesion patch where the cells are separated by a distance $d$ (Fig 5E). The effective free energy of such a doublet configuration can then be written as

$$\mathcal{F}(r_c, h_c, d, P) \approx 2\bar{\gamma}\pi(2r_c^2 + h_c^2) + \pi r_c^2\zeta(d) - 2P(V - V_0) \ , \tag{36}$$

where $V = \pi h_c(3r_c^2 + h_c^2)/6$ is the volume of one cell, $P$ is the cell pressure, and $\zeta(d)$ an effective surface tension at the contact arising from cell-cell adhesion. For a sufficiently large patch compared to the interaction distances $r_{\min}$ and $l$, the effective surface tension can be

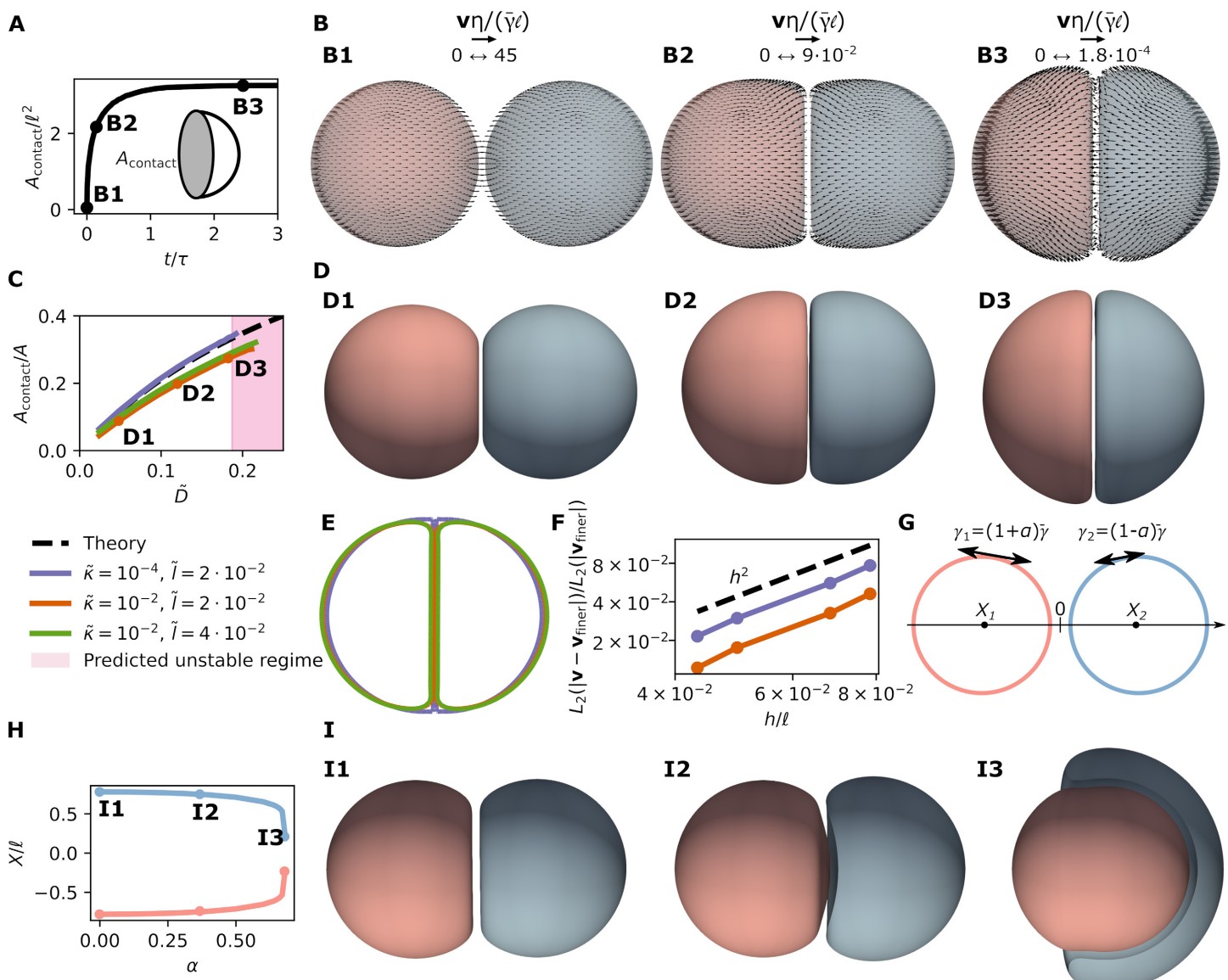

**Fig 5. Dynamics and steady-state shape of a cell doublet.** (A) Normalised area of contact as a function of time ($\tilde{\kappa} = 10^{-2}$, $\tilde{l} = 0.04$). (B) Snapshots of deforming doublet at times indicated in (A), with the cell surface velocity field superimposed. (C) Coloured lines: ratio of contact surface area $A_{\text{contact}}$ to cell surface area $A$, for simulations with different values of $\tilde{l}$ and $\tilde{\kappa}$. Black dotted line: theoretical approximation valid in the limit of $\tilde{\kappa} \to 0$, $\tilde{l} \to 0$, $\tilde{r}_{\min} \to 0$. (D) Snapshots of doublet equilibrium shape for increasing adhesion strength; different parameters in (D1)-(D3) correspond to points labelled in C. (E) Comparison of a slice for the different simulations, with values of $\tilde{l}$ and $\tilde{\kappa}$ indicated in C, and $\tilde{D} = 0.14$. The value of $\tilde{\kappa}$ affects the shape smoothness of the edge of the adhesion patch. (F) Convergence of the method evaluated by computing the $L_2$-norm of the error in the initial velocity field for two different $\tilde{D}$ ($\tilde{D} = 0.05$ (orange) and $\tilde{D} = 0.15$ (blue)), and different average mesh sizes $h$, and comparing the results with a simulation with $h/\ell \approx 2 \cdot 10^{-2}$ (finer). (G) Schematic of adhering doublet, with different active tensions $\gamma^1$ and $\gamma^2$ for each cell. (H) Position of the cell centre of mass $X_1$ and $X_2$, as a function of active tension asymmetry between the two adhering cells, $\alpha = (\gamma^1 - \gamma^2)/(\gamma^1 + \gamma^2)$, where $\gamma^1$ and $\gamma^2$ are the surface tensions of the two cells. Beyond $\alpha = 0.69$, the cell with lowest tension completely engulfs the one with highest tension. (I) Snapshots of doublet equilibrium shape, clipped by a plane passing by the line joining the cell centres, for increasing difference of active surface tension; corresponding to points labelled in H. In H, I: $\tilde{D} = 0.072$, $\tilde{\kappa} = 10^{-2}$, $\tilde{l} = 0.04$.

approximated as (S1 Appendix 9):

$$\zeta(d) = 2\pi \int_0^\infty r dr \; \varphi(\sqrt{r^2 + d^2}) \, , \tag{37}$$

which can be evaluated numerically for a given potential $\varphi$. Minimising the effective free energy (36), one obtains equilibrium values $r_c^*, h_c^*, d^*, P^*$ (S1 Appendix 9), which depend on the effective surface tension of a cell at the contact:

$$\gamma^l = \bar{\gamma}\left(1 - \frac{1}{2}\beta\left(\frac{r_{\min}}{l}\right)\tilde{D}\right) , \tag{38}$$

where the surface tension of the whole interface is $2\gamma^l$. Here, we have introduced a numerical function $\beta\left(\frac{r_{\min}}{l}\right)$, whose functional form depends on the potential $\varphi(r)$. For the value of $r_{\min}/l$ chosen here, $\beta \simeq 10.7$. When the net tension at the contact $2\gamma^l$ becomes negative, for

$$\tilde{D} = \frac{Dr_{\min}l}{\bar{\gamma}} > \frac{2}{\beta} \simeq 0.187 , \tag{39}$$

we expect the system to develop a buckling instability. Indeed, the corresponding threshold for buckling is well predicted by the simulation with smallest $l/\ell$ and bending modulus $\kappa$ (Fig 5C). Before the buckling instability, the ratio between the contact area and the cell surface area as well as the cell pressure $P^*$ are well predicted by the approximate analysis, which become more accurate as $\tilde{\kappa}, \tilde{l} \to 0$ (Fig 5C and S1B Fig).

To further check the numerical method, we analyse the convergence of the velocity field at the initial step as a function of the mesh size $h$ and compare it with the result obtained for a fine mesh $h/\ell \approx 2 \cdot 10^{-2}$, for a fixed value of $\tilde{l} = 0.02$ (Fig 5F). We observe, as in the previous section, convergence of the solution with $\sim h^2$. We also compare the pressure value $P$ in the equilibrium configuration obtained at different values of $h$ and $\tilde{D}$ and observe that here, on average, errors converge as $\sim h^3$ (S1C Fig).

Finally, we examine an asymmetric doublet system where cells have different tensions $\gamma^1$ and $\gamma^2$ (Fig 5G–5I). The corresponding equilibrium state has been considered previously (Ref. [72] and references therein). Fixing a relatively low value of $\tilde{D} = 0.072$, we change the ratio $\alpha = (\gamma^1 - \gamma^2)/(\gamma^1 + \gamma^2)$. As expected from previous studies, we observe that the cell with lowest tension progressively engulfs the cell with highest tension as the ratio $\alpha$ is increased. As a result, their centre of masses approach each other for increasing values of $\alpha$ (Fig 5H and 5I). In our numerical simulations, beyond $\alpha \simeq 0.69$, the cell with lowest tension self-intersects before completely engulfing the one with highest tension (S1D Fig).

## 3.3 Epithelial monolayer

We now discuss simulations of aggregates containing a larger number of cells. We start by considering a sheet of cells with free boundary conditions, intended to resemble the organisation of a simple epithelial island. To obtain an initial condition for such a sheet, spherical cells with homogeneous active tension $\bar{\gamma}$ are positioned with their centres on a 6x6 hexagonal lattice with side $s = 2\ell + r_{\min} + l$. We then let the system relax to its equilibrium state, varying the adhesion parameter $\tilde{D}$, keeping its value below the instability threshold identified in the doublet analysis. To illustrate how the computational load is distributed in a simulation involving several cells, we discuss the computational resources employed in these relaxation simulations in S1 Appendix 11). As the adhesion parameter $\tilde{D}$ is increased, the cellular shapes progressively deviate from loosely adhering spheres to packed columnar cells (Fig 6A and 6B). The reduced volume $v = 6\sqrt{\pi}V/A^{3/2}$ with $V$ the cell volume and $A$ its surface area, a measure of the deviation of the cell shape from a sphere with $v = 1$ being a sphere, is close to 1 for small $\tilde{D}$ and then decreases (Fig 6B).

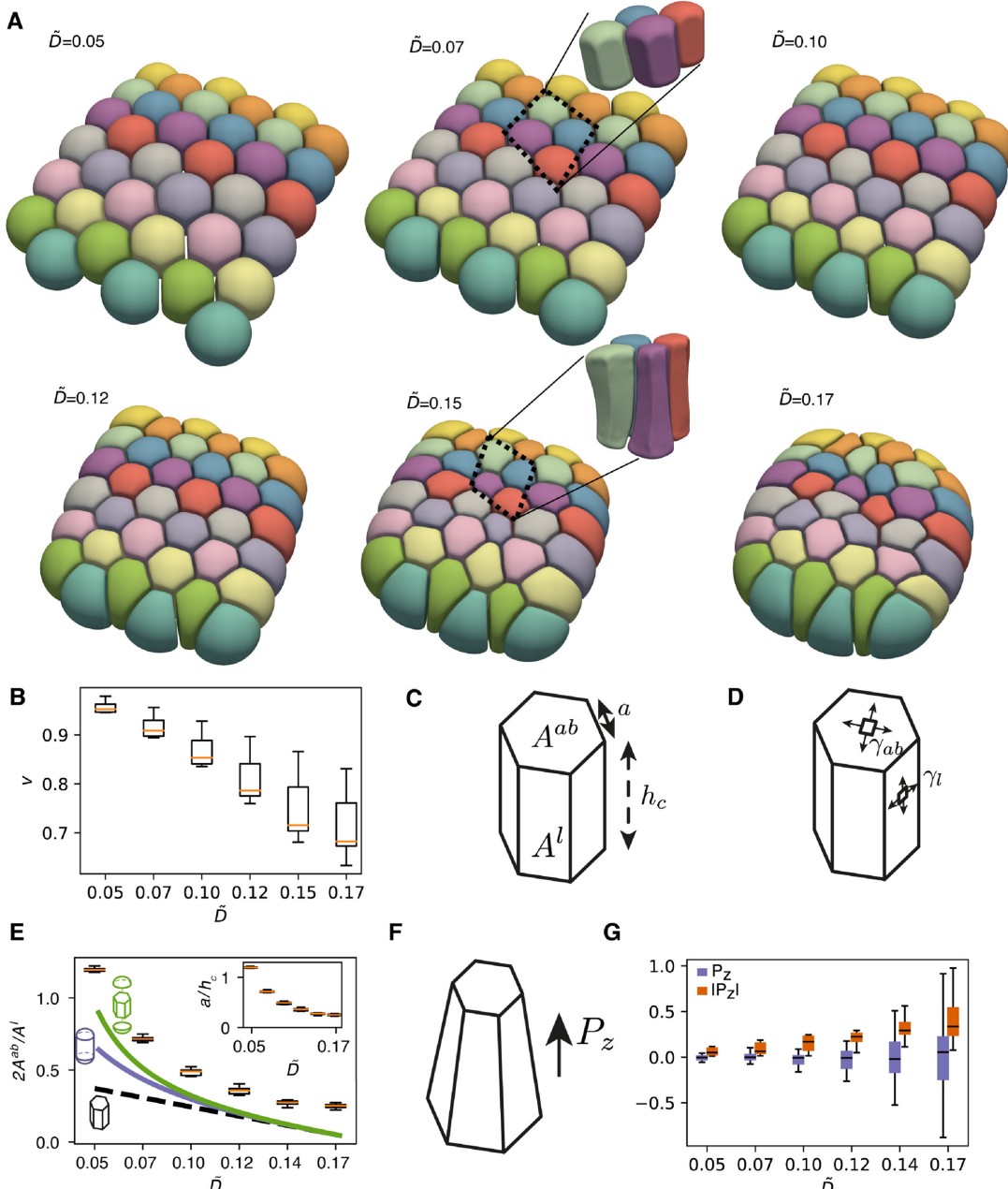

**Fig 6. (A) Simulation results (equilibrium shapes) for a planar sheet for different values of the adhesion parameter**
$\tilde{D} = D r_{min} l / \bar{\gamma}$ ($\tilde{\kappa} = 10^{-2}$, $\tilde{l} = 0.04$). (B) Reduced volume $v$ as a function of $\tilde{D}$. (C) Schematic for the measured apical and basal
cell surface area $A^{ab}$, lateral surface area $A^l$, side length $a$ and cell thickness $h_c$. (D) Results are compared to a simple 3D vertex
model with a lateral surface tension $\gamma_l$ and apical and basal surface tension $\gamma_{ab}$. (E) Box plots: ratio of apico-basal to lateral
surface area $2A^{ab}/A^l$ for the center cells, as a function of the adhesion parameter $\tilde{D}$. Dashed black, blue and green lines:
prediction of simplified theories describing the cell shape as an hexagonal prism, a cylinder with two spherical caps, and the
union of a hexagonal prism with two spherical caps. Inset: cellular aspect ratio $a/h_c$, with $a$ measuring the side of the hexagonal
face and $h_c$ the thickness of the sheet, as a function of $\tilde{D}$. Here $a = \sqrt{2A^{ab}/(3\sqrt{3})}$ and $h_c = A^l/(6a)$. (F) Schematic for the polar
vector $\boldsymbol{P}$ characterising the asymmetry of the cell shape. (G) Box plot for $P_z$ (blue) and $|P_z|$ (red). As adhesion increases, cells
deform asymmetrically in the direction orthogonal to the planar sheet.

We compare the simulation results with a theoretical prediction from a 3D vertex model on a perfect hexagonal lattice, i.e. formed by uniform hexagonal prisms with height $h_c$ and side length $a$ (Fig 6C and 6D). We consider that cells are subjected to an apico-basal tension $\gamma^{ab} = \bar{\gamma}$, lateral tension $\gamma^l = \bar{\gamma}(1 - \beta\tilde{D}/2)$, and an inner pressure $P$ enforcing the cell volume to be equal to the reference volume $V_0 = 4\pi\ell^3/3$. We then write the corresponding effective free energy for a single cell in the tissue (S1 Appendix 10):

$$\mathcal{F}(a, h_c, P) = 2\gamma^{ab}A^{ab} + \gamma^l A^l - P(V - V_0) \ . \tag{40}$$

For a hexagonal prism, $A^{ab} = 3\sqrt{3}a^2/2$ the apical and basal surface area, $A^l = 6ah_c$ the lateral surface area, and the cell volume is $V = 3\sqrt{3}a^2 h_c/2$. Minimising the effective free energy, we obtain the equilibrium ratio $A^{ab}/A^l$ which can be compared to simulation results for the 16 inner cells (Fig 6E). This area ratio is related to the cell aspect ratio, with smaller values corresponding to more columnar cells and larger values corresponding to more squamous cells. Although this simplified model captures the qualitative trend of the cell shape dependency on cell adhesion, it underestimates the simulated area ratio.

We now consider alternative simplified descriptions where (i) each cell is considered as a cylinder connected to two spherical caps, or (ii) use an approximation where the surface area and volume of the cell is defined through the union of a hexagonal prism and two spherical caps (S1 Appendix 10). These choices better capture the actual simulated shapes (Fig 6E), showing that taking the apical and basal curvatures play a significant role in the equilibrium cell shape. These refined models however still underestimate the area ratio $A^{ab}/A^l$. This can arise from the fact that these simplified descriptions do not take into account the surface bending modulus, the tissue-scale deformation due to the system finite size and free boundary conditions, and consider approximate cell shapes.

In addition, we noticed that for higher values of $\tilde{D}$, the cellular shapes appear more heterogeneous (Fig 6A). This heterogeneity appears to be linked to an asymmetry in the cell apical and basal surface areas (see zoomed area comparing how lateral faces look for small $\tilde{D} = 0.07$ and for larger $\tilde{D} = 0.15$ in Fig 6A). To verify this, we introduce a polar order parameter for the shape of cell $I$ (Fig 6F):

$$\boldsymbol{P}_I = \frac{1}{S_I}\int_{\mathcal{S}_I} dS_I \ \boldsymbol{X} - \frac{1}{V_I}\int_{\mathcal{V}_I} dV_I \ \boldsymbol{X}, \tag{41}$$

with $\mathcal{V}_I$ the volumetric domain enclosed by $\mathcal{S}_I$, and $S_I$ and $V_I$ the surface area and volume of cell $I$. We calculate the order parameter for all cells in the simulation and consider its projection on the direction orthogonal to the plane containing the initial cell centers, $z$. We observe that, with increasing adhesion strength $\tilde{D}$, the average projected cell polarity does not clearly deviate from zero, $\langle P_z \rangle \simeq 0$, but the average of the absolute value of the projected cell polarity, $\langle |P_z| \rangle$, strongly increases (Fig 6G). This suggests that at high enough adhesion, cells adopt polarised apico-basal shapes orthogonal to the plane of the tissue, with no consistent overall shape polarisation orthogonal to the tissue (Fig 6A). Possibly, such a spatial arrangement favours larger contact areas, which is beneficial at large adhesion.

## 3.4 Adding cell divisions: Growth of an organoid

We now discuss simulations modelling the growth of a cell aggregate from a single cell, for which we introduce cell divisions in our framework. When a cell divides, the mother cell is replaced by two daughter cells as follows: a randomly oriented plane passing through the mother cell centre is selected, splitting the mother cell in two parts. The two daughter cells are

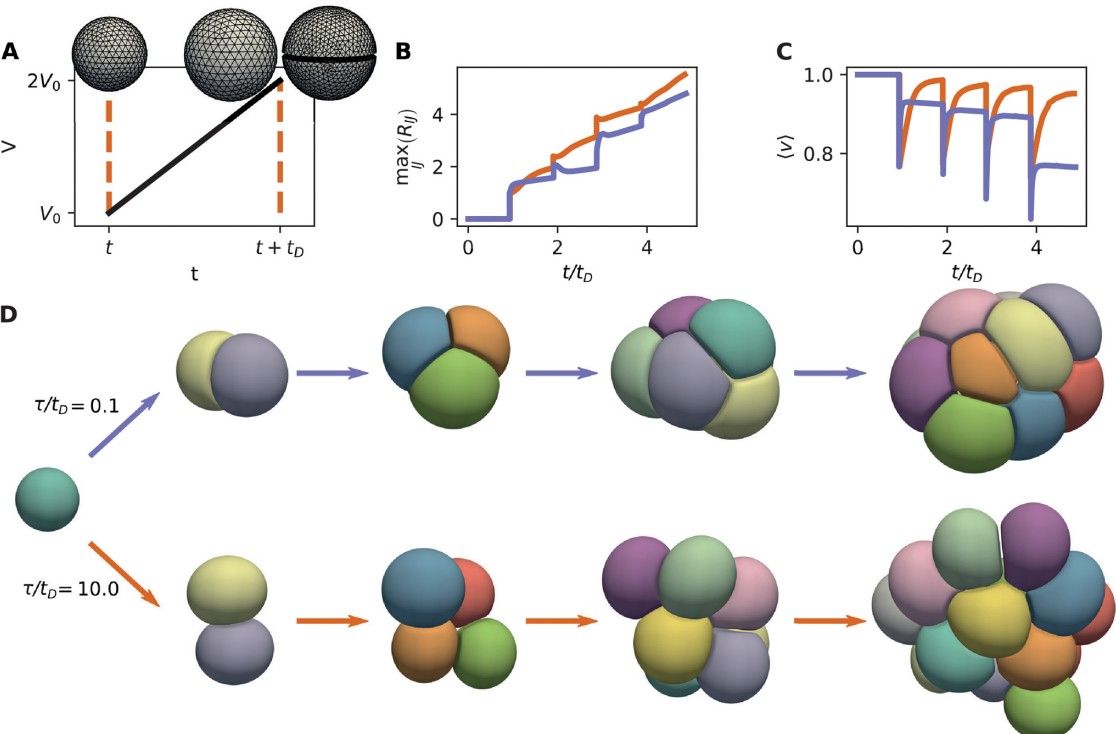

**Fig 7. Growth of small cell aggregates, driven by synchronous cell divisions and cell volume increase ($\tilde{D} = 0.15$, $\tilde{\kappa} = 10^{-2}$, $\tilde{\ell} = 0.04$).** (A) Each cell doubles its volume between birth and division, over a cell cycle time $t_D$. Cell division is introduced by splitting the mother cell with a plane passing through the cell centre, and generating two daughter cells separated by a distance $d^*$ ($d^*/2$ from the division plane). (B-D) Simulation results for two values of the ratio $\tau/t_D$. (B): Largest centre-to-centre cell distance $\max_{\langle I,J \rangle} R_{IJ}$, as a function of time. Jumps correspond to cell division events. (C) Average reduced volume $\langle v \rangle$ as a function of time. (D) Snapshots of simulations of two growing aggregates.

then separated by this plane and simply fill the original shape of the mother, except for a small region that separates the daughter cells by a distance $d^*$, perpendicular to the division plane (Fig 7A). To determine timepoints of cell division, each cell is assigned a cell cycle time $t_D$, which we take equal for all cells.

We assume that in between divisions, cell volume follows a linear growth law $\dot{V}^I = V_0/t_D$, where $V_0$ is the volume of the cell at its birth. This effectively leads to cells doubling their volume during their lifetime—we note however that a small volume loss occurs at division due to the initial separation of the daughter cells by a distance $d^*$ (Fig 7A). At each time point, cell volume is imposed through the Lagrange multiplier $P_I$.

We then simulate the growth of an aggregate starting from a single cell (Fig 7B–7D, S2 and S3 Videos). The dynamics of the growing aggregate strongly depends on $\tau/t_D$, which measures the ratio between a characteristic time scale of cell shape relaxation, and the cell cycle time.

For smaller values of $\tau/t_D$, the growing aggregate is more compact (Fig 7B), cells have a smaller reduced volume and have therefore shapes further away from spheres (Fig 7C). Here, the aggregate compactness is measured by calculating the maximum distance between cell centres (Fig 7B). These observations indicate that the shape of a cell aggregate can strongly depend on a competition between its growth rate and internal mechanical relaxation times.

## 4 Discussion

The framework of interacting active surfaces introduced here is a novel method to study the mechanics of cell aggregates such as early developing embryos or organoids, and opens the door to their systematic modelling and simulation. We have demonstrated here that it can be used to study in detail the shape of adhering cell doublets, simple epithelia, as well as growing cellular aggregates. Our method is well-suited to capture the mechanics of tissues and organoids connecting it to cell level processes such as cortical flows, cortical tension and cellular adhesion in the organisation of a cellular aggregate.

In this study we have restricted ourselves to relatively simple constitutive equations for the tension and bending moment tensors (Eq (3)), and we have considered situations with a uniform and constant surface tension within each cell. Our method is based on using the virtual work principle (Eq (2)), a very general statement of force and torque balance for a surface, to obtain a set of algebraic equations for the cell surface described with finite elements. As such it is versatile and we expect that more complex constitutive equations, corresponding to more detailed physical descriptions of the cell surface, can be easily introduced in our description. We now discuss some of these possible extensions of our framework.

We have not included here apico-basal polarity, an axis of cell organisation which results from a spatially segregated protein distribution and inhomogeneous cytoskeletal structures [73]. To take this into account, one could introduce a polarity field in each cell and consider an active tension $\gamma$ on the cell surface whose value at each point depends on the polarity field orientation. This could be used to introduce, for instance, differences in apical, basal and lateral surface tension which are taken into account in 3D vertex models [19].

It would be natural to introduce a concentration field on the cell surface, describing a regulator of the cortical tension, such as myosin concentration. At the level of a single surface, such coupling between cortical flows and its regulator can give rise to pattern formation, spontaneous symmetry breaking and shape oscillation [40, 74, 75]. The dynamics of the concentration per unit area, $c$, of such a regulator can be obtained from the balance equation on the surface:

$$D_t c + c v_i^i + \nabla \cdot \boldsymbol{j} = r, \tag{42}$$

where $D_t c$ is the material derivative of $c$ ($\partial_t c$ in a Lagrangian description), $\boldsymbol{j}$ is the flux of $c$ relative to the centre of mass, and $r$ is a reaction rate. A natural choice for the flux would be $\boldsymbol{j} = -D\nabla c$ to represent diffusion according to Fick's law. A natural choice for the reaction rate would be $r = k_{\mathrm{on}} - k_{\mathrm{off}} c$ for turnover dynamics, with target concentration $c_0 = k_{\mathrm{on}}/k_{\mathrm{off}}$ and typical turnover time $\tau = k_{\mathrm{off}}^{-1}$. The discretisation of such fields can be easily introduced in our framework, following the methods detailed in [41, 76]. One could then model the effect of this concentration on the active tension by assuming e.g. $\gamma(c) = \gamma_0 c/c_0$ with $\gamma_0$ a reference tension at $c = c_0$.

Importantly, the model introduced here is based on an interaction potential between cells, which can be motivated microscopically from a description of cell-cell linkers that equilibrate quickly to their Boltzmann distribution, with free linkers on the surface in contact with a reservoir imposing a constant concentration. From a computational perspective, the exact integration of the interaction potential requires the computation of double integrals, which have a large computational cost. Alternatively, one could approximate the double integrals further in the limit $\tilde{l}, \tilde{r}_{\min} \ll 1$ by considering the interaction of each point on surface $\mathcal{S}_I$ only with its closest point projection on $\mathcal{S}_J$, following classical numerical approaches for the adhesion between interfaces (Ref. [77] and references therein).

On the other hand, the interplay of adhesive cell-cell linkers such as E-cadherin with cortical dynamics also plays an important role in orchestrating cell adhesion [78]. Unlike the

specific adhesion of solid interfaces, cell-cell adhesion dynamics involves a complex interplay between the diffusion, advection and binding dynamics of linkers [79]. Notably, E-cadherin junctions have been shown to be mechanosensitive [80] and to act to regulate the actomyosin levels at junctions [81]. To take these effects into account, an explicit description of E-cadherin concentration on the cell surface might be required. Thus, an extension of our model could introduce explicitly two-point density fields $c_{IJ}(\boldsymbol{X}_I, \boldsymbol{X}_J)$ representing the concentration of bound linkers between cells $I$ and $J$, as well as a concentration field of free linkers on each cell $c_I$. Alternatively, one could introduce cell-cell adhesion by considering a finite number of explicitly described individual linkers [82].

Our model does not account for the friction generated by relative surface flows between cells that adhere to each other, which is likely to play an important role during cell rearrangements. The effective friction stemming from an ensemble of transiently binding and unbinding linkers can be modelled effectively with a friction coefficient motivated by microscopic models such as a Lacker-Peskin model [83], which lead to predictions of force-velocity relations which depend on whether linkers are force-sensitive, e.g. slip or catch bonds [84, 85]. One could include these terms systematically in our finite element discretisation following the ideas in [86, 87].

In its current version and with these additions, we hope that the interacting active surface framework will be a useful tool to investigate the mechanics and self-organisation of cellular aggregates.

## Supporting information

**S1 Appendix. Details of derivations and computational framework used in the main text.**
(PDF)

**S1 Fig.** (A) Side view, cut by a plane perpendicular to the adhesion patch of an adhering doublet for $\tilde{D}$ larger than the critical value. The system develops a buckling instability that grows with time. The instability eventually leads to self-intersections (right-most image, where the blue cell has collapsed). (B) Coloured lines: pressure for simulations with different values of $\tilde{l}$ and $\tilde{\kappa}$. Black dotted line: theoretical approximation valid in the limit of $\tilde{\kappa} \to 0, \tilde{l} \to 0, \tilde{r}_{\min} \to 0$. (C) Convergence of the method evaluated by computing the inner cell pressure $P$ for different average mesh sizes $h$, and comparing the results with a simulation with $h/\ell \approx 2 \cdot 10^{-2}$ (finer). For each $h$, we compute a box plot using different values of $\tilde{D}$ and fixed $\tilde{\kappa} = 10^{-2}, \tilde{l} = 0.02$. (D) Side view, cut by a plane perpendicular to the adhesion patch of an adhering doublet with asymmetric tension for $\alpha = 0.7$; the cell with lower tension (blue) engulfing the cell with higher tension (red) develops a self-intersection in our numerical simulations (right-most image).
(PDF)

**S1 Video. Comparison of the mesh relaxation method for different values of the tolerance for normal motion (left column: $10^{-3}$, right column: $10^{-4}$) and different mesh sizes (top row: $4 \times 10^{-2}$, bottom row: $2 \times 10^{-2}$).** The colormap represents the distance to the original surface, which is in all cases below $5 \times 10^{-3}$.
(M4V)

**S2 Video. Growth of a cell aggregate driven by cell growth and cell divisions in the case where the cell lifetime $t_D$ is ten times larger than the shape relaxation time-scale $\tau$.**
(MP4)

**S3 Video. Growth of a cell aggregate driven by cell growth and cell divisions in the case where the cell lifetime $t_D$ is ten times smaller than the shape relaxation time-scale $\tau$.** (MP4)

## Acknowledgments

We thank Quentin Vagne and Guillermo Vilanova for comments on the manuscript.

## Author Contributions

**Conceptualization:** Alejandro Torres-Sánchez, Guillaume Salbreux.

**Funding acquisition:** Guillaume Salbreux.

**Methodology:** Alejandro Torres-Sánchez, Guillaume Salbreux.

**Software:** Alejandro Torres-Sánchez.

**Supervision:** Guillaume Salbreux.

**Validation:** Max Kerr Winter.

**Writing – original draft:** Alejandro Torres-Sánchez, Guillaume Salbreux.

**Writing – review & editing:** Alejandro Torres-Sánchez, Guillaume Salbreux.

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
