## [Decision Letter · Decision Letter 0]

8 Jul 2022

Dear Dr Salbreux,

Thank you very much for submitting your manuscript "Interacting active surfaces: a model for three-dimensional cell aggregates" for consideration at PLOS Computational Biology.

As with all papers reviewed by the journal, your manuscript was reviewed by members of the editorial board and by several independent reviewers. In light of the reviews (below this email), we would like to invite the resubmission of a significantly-revised version that takes into account the reviewers' comments.

We have now finally received the three reports for your manuscript "Interacting active surfaces: a model for three-dimensional cell aggregates". As you can see, all the Reviewers agree in the importance and interest of the manuscript, and it most definitely would make a great contribution to PLOS Computational Biology. However, the Reviewers have some comments, very specially Reviewer 1. Before the manuscript can be accepted, these comments have to be addressed.

We cannot make any decision about publication until we have seen the revised manuscript and your response to the reviewers' comments. Your revised manuscript is also likely to be sent to reviewers for further evaluation.

Sincerely,

Saúl Ares, Ph.D.

Guest Editor

PLOS Computational Biology

Douglas Lauffenburger

Deputy Editor

PLOS Computational Biology

Dear Guillaume,

We have now finally received the three reports for your manuscript "Interacting active surfaces: a model for three-dimensional cell aggregates". As you can see, all the Reviewers agree in the importance and interest of the manuscript, and it most definitely would make a great contribution to PLOS Computational Biology. However, the Reviewers have some comments, very specially Reviewer 1. Before the manuscript can be accepted, these comments have to be addressed. Please let me know if I can be of any assistance during the process.

Best regards,

Saúl

Reviewer's Responses to Questions

**Comments to the Authors:**

Reviewer #1: Please see my attached comments (review.pdf)

Reviewer #2: PCOMPBIOL-D-22-00456

Interacting active surfaces: a model for three-dimensional cell aggregates

The interest in modeling biological surfaces is not recent, but most studies mainly focused on passive membranes. More recently, several papers extended the study of fluid surfaces to active surfaces, which is a natural and necessary step to making better models with the hope of quantitatively confronting theory with biological experiments.

This article provides significant advances for the simulation of three-dimensional biological systems that will undoubtedly positively impact the field. The authors put forward a numerical approach capable of handling multicellular aggregates with the dynamical evolution of surface fields. There is a good balance in the article between theory (where there is nothing particularly new, but this is not the article's focus), numerics, and results. The cell mechanics is dictated here by the actomyosin cortex at a long (viscous) timescale, and cell-cell adhesion is modeled using the phenomenological Morse potential. They present biologically relevant applications of the numerical scheme. The paper is well written and would appeal to what I consider typical readers of PLOS. After some minor revisions, I would be happy to recommend it to the editors of PLOS for publication.

Introduction:

In the introduction, the reader might benefit from a brief discussion on the physical/biological motivation on why only the cortex is necessary for describing the cell dynamics and why the membrane plays no role.

The authors say on pg. 2, "In this framework, topological transitions appear as a natural output of the remodeling of cell-cell interactions and are not treated explicitly." However, it is not clear how the topological transitions will happen naturally. This is particularly relevant because Sec 3.4 deals with cell divisions. Still, the authors chose to cut the cell in a random plane instead of modeling the division dynamics (as in the ref. HB da Rocha, J Bleyer, H Turlier - Journal of the Mechanics and Physics of Solids, 2022, although I believe that here the authors did not treat topological transitions).

Materials and methods

Pg. 4:

I'm curious to know why the authors chose the virtual work principle to formulate what is clearly a fluid problem. Wouldn't the Onsager variational principle or virtual power principle be more relevant here?

I believe that because they deal essentially with stokes flow, ultimately, there will be no problem.

Pg. 8:

The section on the forces arising from cell-cell interactions is presently a bit confusing. The authors start by stating a general function for the cell-cell interactions, then go on to motivate a particular form for the cell-cell interaction energy considering stretchable linkers connecting the cells with binding and unbinding dynamics. Then, the authors observe that such potential fails to capture repulsion and turns to a phenomenological potential, the Morse potential. I believe that this section could go direct to the point.

Pg. 11:

The authors devote a subsection to presenting an already established surface reparametrization method. Yet, the exposition is restricted to a single cell without discussing the reparametrization in the numerical experiments. Thus, I would suggest the authors either shorten the exposition to a minimum or complete it with (1) information about the extension to cell aggregates and (2) error assessment in the numerical examples to give the reader an idea of the accuracy of the method. For instance, are there any difficulties when parametrizing cell aggregates (e.g., possible self-intersections)? What is the reparametrization error in terms of volume conservation?

Pg. 20:

In the discussion section, the authors report that the exact integration of the interaction potential has a high computational cost. I think the reader would also benefit from knowing the relative weight of this simulation phase, i.e., the percentage of time devoted to integrating the interaction potential with respect to the total simulation time.

Finally, the authors highlight in the abstract and the introduction their hybrid MPI-OpenMP parallelized C++ implementation. Yet, the reader is given very few details in the subsequent exposition on the parallelization, especially in the numerical experiments. I would suggest reporting at least minimal quantities to give the reader an idea of the parallel performance of the code. For instance, to inform about global problem sizes, max. local problem sizes, max. number of cores employed, total simulation times, and CPU specification for the largest numerical examples.

Reviewer #3: This work presents a general computational model for the interaction of active fluidic surfaces. The manuscript is very well written, interesting and important. The presented examples are well chosen and clearly demonstrate the versatility of the proposed model. The work is suitable for publication provided the following comments are addressed:

1. Sec. 2.1: Usually the exact opposite notation is used: Greek indices for surface coordinates and Latin indices for 3D coordinates. The authors should motivate their choice if it is non-standard.

2. Sec. 2.1: g_ij, C_ij, etc. are not tensors (even if many authors erroneously call them so). They are just components, possibly of tensors. The full tensor includes the basis vectors, which makes a big difference, esp. for curvilinear coordinates.

3. Following Eq. (7): The authors should note here that the scope of constitutive laws for fluidic membranes as it stems from irreversible thermodynamics has also been discussed in Sahu et al. (2017), “Irreversible thermodynamics of curved lipid membranes”, Phys. Rev. E, 96:042409

4. Sec. 3.1: It is not clear to me, whether the surface shape changes in this example. There is an out-of-plane surface velocity but then the analytical solution seems to be based on a spherical shape.

5. It is not clear to me to what degree the surface flow has an influence on the presented results in Sec. 3.2. & 3.3. (and also 3.4). Did the authors vary the relevant parameters to examine? It would be interesting to see this.

6. At some point the authors mention that the active surface tension is considered a given constant, but in principle could depend on some concentration and vary locally. Is this the case in all four examples? The authors should clarify.

**Have the authors made all data and (if applicable) computational code underlying the findings in their manuscript fully available?**

Reviewer #1: Yes

Reviewer #2: Yes

Reviewer #3: None

PLOS authors have the option to publish the peer review history of their article (what does this mean?). If published, this will include your full peer review and any attached files.

Reviewer #1: No

Reviewer #2: No

Reviewer #3: No
---

## [Decision Letter · Decision Letter 1]

7 Nov 2022

Dear Dr. Salbreux,

Thank you very much for submitting your manuscript "Interacting active surfaces: a model for three-dimensional cell aggregates" for consideration at PLOS Computational Biology. As with all papers reviewed by the journal, your manuscript was reviewed by members of the editorial board and by several independent reviewers. The reviewers appreciated the attention to an important topic. Based on the reviews, we are likely to accept this manuscript for publication, providing that you modify the manuscript according to the review recommendations.

We have now received all the Reviewer's reports. While two Reviewers recommend publication without more comments, one Reviewer still raises some relevant points. I think that clarifying these points will be positive for the manuscript, and therefore I recommend to address them carefully.

Best regards,

Saúl Ares

Sincerely,

Saúl Ares, Ph.D.

Guest Editor

PLOS Computational Biology

Douglas Lauffenburger

Section Editor

PLOS Computational Biology

Dear Dr. Salbreux,

We have now received all the Reviewer's reports. While two Reviewers recommend publication without more comments, one Reviewer still raises some relevant points. I think that clarifying these points will be positive for the manuscript, and therefore I recommend to address them carefully.

Best regards,

Saúl Ares

Reviewer's Responses to Questions

**Comments to the Authors:**

Reviewer #1: I thank the authors for their careful responses to my comments, as well as the comments of the other reviewers. I am satisfied with the response to my minor concerns, as well as two of my three major concerns. However, I remain worried about your treatment of the active tension, in which $\\gamma$ is either a prescribed constant or a prescribed field (which does not vary in time). I now understand that you seek to describe a cell in the limit where cytoskeletal elements are constantly and rapidly turning over, such that local inhomogeneities are quickly removed and can be neglected on the natural timescale of membrane deformations/flows. In my mind, I picture a reservoir of cytoskeletal elements in the cell cytoplasm to allow such a situation. However, this brings up additional questions:

1. It seems such a composite system (membrane + cortex) is no longer described by the Helfrich free energy, which is appropriate for a two-dimensional system with in-plane fluidity. Given your statement that cortical stresses dominate membrane ones, wouldn't an energy density for a material with in-plane elasticity (and not fluidity) and out-of-plane bending be more appropriate? Your edits to the introduction say that the composite system is in-plane fluid on the timescale of tens of seconds, but many cellular phenomena of interest happen on shorter timescales. Wouldn't your choice of free energy then imply you cannot resolve any of those?

2. Generally, a bending deformation costs energy because one side of the sheet is stretched while the other is compressed, both of which are energetically unfavorable. However, if the turnover of cytoskeletal elements is so rapid, wouldn't this bending energy no longer exist, since there is no way for stress to be built up? Rather, the actomyosin cortex would always remain in a stress-free configuration. It seems the cortex turnover would have to in fact be slow relative to shape changes in order for bending forces to arise.

3. Even under the assumption that a rapid turnover prevents any density changes of the cortex, and that the Helfrich energy remains appropriate, wouldn't the lipid membrane itself still undergo density variations as the composite system deforms? In this case, even if the density of the cortex is constant, there will still be surface tension gradients caused by the membrane. Moreover, since the gradient of the "cortex" surface tension is zero, it is these membrane surface tensions which will dominate in the in-plane equations. A mass balance would then be required in your formulation.

4. I believe your response regarding the Rayleigh--Plateau instability is incorrect. It is true that pearling arises in a cylindrical fluid column with constant surface tension. However, here the driving mechanism is the development of pressure gradients in the fluid column in response to an initial perturbation. These pressure gradients are due to the Young--Laplace equation (constant tension and curvature gradients lead to pressure gradients), and drive a viscous fluid flow (via the Stokes equation) which reinforces the initial disturbance. However, since your numerical method does not capture any pressure gradients in the surrounding medium, I believe it cannot describe such an instability. If I am correct, then the inability of your equations to describe such a well-known and commonly observed cellular phenomena would be very troubling, and would further confirm my belief that you need an explicit statement of local mass balance in your dynamics.

Reviewer #2: I have reviewed the manuscript's revised version and the author's response to the previous comments. The authors made an effort to convince the referees about the merits of their work and revised the manuscript according to the referees' questions and comments.

On a minor note, I agree with Reviewer #3 point 1: using Latin indices to represent surfaces is disturbing and might confuse readers used to the classic notation (it confused me at first sight). Although I believe uniformizing the notation is helpful for overall readability, it does not remove the merit of the work.

Given this, I recommend the publication of the revised version in PLOS Computational Biology.

Reviewer #3: The authors have accounted for all my comments, and as far as I can see, also the other reviewer's comments. In my mind the work is suitable for publication now.

**Have the authors made all data and (if applicable) computational code underlying the findings in their manuscript fully available?**

Reviewer #1: Yes

Reviewer #2: Yes

Reviewer #3: Yes

PLOS authors have the option to publish the peer review history of their article (what does this mean?). If published, this will include your full peer review and any attached files.

Reviewer #1: No

Reviewer #2: No

Reviewer #3: No

Figure Files:

Data Requirements:

Reproducibility:

References:

---

## [Editor Report · Decision Letter 2]

26 Nov 2022

Dear Dr. Salbreux,

We are pleased to inform you that your manuscript 'Interacting active surfaces: a model for three-dimensional cell aggregates' has been provisionally accepted for publication in PLOS Computational Biology.

Sincerely,

Saúl Ares, Ph.D.

Guest Editor

PLOS Computational Biology

Douglas Lauffenburger

Section Editor

PLOS Computational Biology

Dear Dr. Salbreux,

I believe the discussion with the Reviewers has helped to make the manuscript clearer and all the assumptions more explicit. I find your last response and version satisfactory, and see no gain in any further discussion. Therefore, your manuscript is accepted in its current form.

---

## [Editor Report · Acceptance letter]

9 Dec 2022

PCOMPBIOL-D-22-00456R2 

Interacting active surfaces: a model for three-dimensional cell aggregates

Dear Dr Salbreux,

I am pleased to inform you that your manuscript has been formally accepted for publication in PLOS Computational Biology. Your manuscript is now with our production department and you will be notified of the publication date in due course.

With kind regards,

Zsofia Freund
